# Chitosan Catalyzed Novel Piperidinium Dicoumarol: Green Synthesis, X-ray Diffraction, Hirshfeld Surface and DFT Studies

**DOI:** 10.3390/polym14091854

**Published:** 2022-04-30

**Authors:** Mohammad Asad, Muhammad Nadeem Arshad, Mohammed Musthafa T.N., Abdullah M. Asiri

**Affiliations:** 1Center of Excellence for Advanced Materials Research (CEAMR), King Abdulaziz University, P.O. Box 80203, Jeddah 21589, Saudi Arabia; mnachemist@hotmail.com (M.N.A.); aasiri2@kau.edu.sa (A.M.A.); 2Chemistry Department, Faculty of Science, King Abdulaziz University, P.O. Box 80203, Jeddah 21589, Saudi Arabia; 3Research & Postgraduate Department of Chemistry, MES Kalladi College, Mannarkkad, 678583 (Affiliated to University of Calicut), Kerala, India; drmusthafa@meskc.ac.in

**Keywords:** 3-formylchromone, 4-hydroxycoumarin, single crystal X-ray diffraction, Hirshfeld surface analysis, DFT study

## Abstract

The novel piperidinium dicoumarol has been synthesized by the reaction of 3-formylchromone, 4-hydroxycoumarin, and piperidine under chitosan catalyzed solvent-free green conditions. FT-IR and NMR spectroscopy established the structure of dicoumarol, which was further confirmed by a single X-ray diffraction study. The single diffraction study has revealed the hydrogen bonding interactions, which were further validated by Hirshfeld surface analysis. Geometry optimizations of dicoumarol have been performed at the DFT level of theory by the B3LYP acting along with Gaussian 16, revision B.01 to calculate the geometric and electronic structure parameters.

## 1. Introduction

Oxygen-containing coumarin and chromone heterocycles are well-known moieties due to their nucleophilic character, and the presence of a C=C bond in them gives rise to dicumarols and other potential compounds by reaction with different functional groups [1,2,3]. The various substituted coumarins and chromones have also been reacted with aldehydes, ketones, amines, and active methylene compounds to form highly functionalized heterocyclic derivatives with significant biological spectrum [4,5]. The simple, conventional and one-pot synthesis has been applied for the preparation of coumarin and chromone derivatives. These heterocyclic derivatives exhibit a potential role in medical innovation over the last few decades of observation [6,7]. Many of the heterocyclic compounds and their derivatives are utilized as new lead molecules for the discovery of drugs with different pharmaceutical significance [8]. The development of novel functional compounds with efficient bioactivities via cheap and short synthetic methodologies is an important task in organic synthesis and pharmaceutical chemistry [9,10].

Biscoumarins, which are bridge-substituted dimers of 4-hydroxycoumarin, have also been discovered to have potent phytochemical properties, HIV-1 integrase inhibitors, antimicrobial and cytotoxic activities [11]. As a result, a large variety of methods for the synthesis of biscoumarin derivatives have previously been published. However, the incorporation of the piperidine moiety into the biscoumarin units has not been documented in the literature. Hence, we planned to prepare piperidinium monohydrate of dicoumarol, which may be expected to have better biological and industrial applications. For many years, efforts in the development of cleaner, more sustainable chemistry, as well as the discovery of heterogeneous catalysts, have been a key focus of research. These can be split into two categories: polymer-based and inorganic-based materials [12,13]. Most recently, the use of biopolymers as polymeric catalysts as well as catalyst supports has played an important role in the area of chemical research. Starch and cellulose are advantageous examples, that are widely employed in a variety of applications. Similarly, chitin and its derivative chitosan are used in a variety of applications, including as catalysts and catalyst supports. Because of its chemophysical and biological features, such as being hydrophilic, positively charged, biodegradable, non-toxic, and biocompatible, chitosan is a recognized biopolymer. Thus, it is a good heterogeneous catalyst and catalytic supporting medium because of its free -NH_2_ groups and hydrogen bonding [14]. Additionally, solvent-free reactions and catalysis provide an important opportunity to achieve the goals of green chemistry [15]. Solvent-free organic synthesis is typically faster, higher yielding with cleaner products, reducing energy consumption and reaction time, which is environmentally benign and involves easy and mild operational procedures in comparison to the classical reactions [16].

In search of more efficient heterocycles, herein we report the simple and convenient synthesis of piperidinium dicoumarol, by the reaction of 3-formylchromone and 4-hydroxycoumarin via chitosan catalyzed green conditions with good yield. The novel dicoumarol was characterized based on elemental and spectroscopic analysis and was additionally corroborated using a single-crystal X-ray diffraction analysis. The single diffraction study has revealed the hydrogen bonding interactions, which were further explored by the Hirshfeld study and by a combined experimental and theoretical DFT study.

## 2. Materials and Methods

### 2.1. General

Chemicals were collected from Sigma-Aldrich chemical distributors. The Stuart Scientific SMP3, version 5.0 melting point apparatus (Bibby Scientific Limited, Staffordshire, UK) was used to measure melting points, which were reported as uncorrected. On a Thermo Scientific NICOLET iS50 FT-IR spectrometer, the FT-IR spectra were recorded as clean (Thermo Scientific, Madison, WI, USA). ^1^H and ^13^C NMR spectra were collected in CDCl_3_ using TMS as an internal standard on a Bruker 850 MHz instrument. Chemical shifts are expressed in ppm while coupling constants are expressed in hertz. For TLC, silica gel pre-coated aluminum sheets (Type60 GF254, Merck) were used, and spots were detected using a UV-lamp set to 254 nm or 360 nm.

### 2.2. Synthesis of Piperidinium 3-[(4-Hydroxy-2-oxo-2H-chromen-3-yl)-(4-oxo-4H-chromen-3-yl)-methyl]-2-oxo-2H-chromen-4-olate Monohydrate (***4***) under Different Solvent Assisted Conditions Using Chitosan as Catalyst

To the mixture of 3-formylchromone (1 g, 0.0057 moles), 4-hydroxycoumarin (1.86 g, 0.0114 moles), piperidine (567.5 µL, 0.0057 mole) in different solvents, such as PEG-400, ethanol, methanol, water, and chitosan (100 mg) was added to it. The reaction mixture was refluxed for a particular period of time. After the reaction was completed, the mixture was allowed to cool once, and the resulting mixture was extracted with hot ethyl acetate (10 mL). The solid catalyst obtained was filtered, washed and air-dried. The filtrate was concentrated and recrystallized from a chloroform-methanol mixture to afford light yellow crystals of **4**.

### 2.3. Chitosan Catalyzed Green Synthesis of Piperidinium 3-[(4-Hydroxy-2-oxo-2H-chromen-3-yl)-(4-oxo-4H-chromen-3-yl)-methyl]-2-oxo-2H-chromen-4-olate Monohydrate (***4***)

To the mixture of 3-formylchromone (1 g, 0.0057 moles), 4-hydroxycoumarin (1.86 g, 0.0114 moles) and piperidine (567.5 µL, 0.0057 moles) taken in a mortar, 100 mg of chitosan were added. The reaction mixture was then mixed well using a pestle and then transferred into a 100 mL beaker. It is then heated over a hot plate at 70 °C. TLC was used to monitor the reaction’s progress. The mixture was allowed to cool once the reaction was completed, and was extracted with hot ethyl acetate (10 mL). The catalyst was filtered and obtained as a solid, washed, and dried at room temperature. The filtrate was concentrated and recrystallized from a chloroform-methanol mixture to afford light yellow crystals of **4**.

### 2.4. Recycling and Reuse of Chitosan Catalyst for the Synthesis of Piperidinium 3-[(4-Hydroxy-2-oxo-2H-chromen-3-yl)-(4-oxo-4H-chromen-3-yl)-methyl]-2-oxo-2H-chromen-4-olate Monohydrate (***4***)

After completion of the reaction (checked by TLC) the product was isolated from the reaction mixture by adding 10 mL of hot ethyl acetate into the reaction mixture. The insoluble catalyst in hot ethyl acetate was removed by filtration, washed with ethyl acetate, and dried. The recovered catalyst was further used for the next cycles. The filtrate was concentrated and recrystallized from a chloroform-methanol mixture to afford light yellow crystals of **4**.

### 2.5. Characterization Data of Piperidinium 3-[(4-Hydroxy-2-oxo-2H-chromen-3-yl)-(4-oxo-4H-chromen-3-yl)-methyl]-2-oxo-2H-chromen-4-olate Monohydrate (***4***)

M.p. 304–305 °C; IR (neat) v_max_: 3412 (OH, coumarin), 1680 (C=O, coumarin), 1642 (C=O, chromone), 1575 (C=C, coumarin) cm^−1^; ^1^HNMR (850 MHz, CDCl_3_): δ/ppm 1.64 (1H, m, CH-chiral), 1.70(2H, m, CH_2_-piperidinium), 1.946(4H, m, CH_2_-piperidinium), 3.277 (4H, s, 2CH_2_-piperidinium), 3.594(1H, t, NH_2_), 3.676(1H, t, NH_2_), 7.398(4H, m, Aromatic), 7.470(2H, d, Aromatic), 7.65(2H, d, Aromatic), 7.77(1H, m, Aromatic), 8.06(4H, m, Aromatic), 9.158(1H, s, O-H); ^13^C NMR (213.7 MHz, CDCl_3_): δ/ppm 15, 31.3, 2(31.9), 2(49.7), 92, 116.3, 118.1, 120.6, 122.8, 2(124.9), 2(127.1), 2(128.2), 129.8, 134.7, 137.6, 149.4, 2(151.3), 158.2, 161.9, 166.3, 186.8. Mass: *m*/*z* 583 (M^+^); Anal. Calcd for C_33_H_29_NO_9_: C, 67.92; H, 5.01; N, 2.40; Found: C, 67.64, H, 5.22, N, 2.54.

### 2.6. Single Crystal X-ray Crystallography

The synthesized material is a zwitterion compound having negatively and positively charged moieties along with a water molecule. The compound was crystallized and screened under a microscope, and the data were collected using an Agilent SuperNova (Dual-source) Agilent Technologies Diffractometer with microfocus Cu/Mo K radiation. The data-gathering program was CrysAlisPro [17], the system temperature was 296 K, and the radiation source was MoK radiation. The structures were solved and resolved using SHELXS–97 and the direct approach [18], and then refined using SHELXL–97 and full-matrix least-squares methods on F2. WinGX [19] came pre-installed with SHELXL–97. Full-matrix least-squares approaches were used to tune all non–hydrogen atoms anisotropically [18]. The figures were created using well-known tools, such as PLATON [20] and ORTEP [21].

Where the C–H distance is 0.93 and Uiso(H) = 1.2 Ueq(C) for carbon atoms, all aromatic hydrogen atoms were geometrically identified and considered as riding atoms. Similarly, methylene hydrogen atoms were positioned geometrically with *d*C-H=0.97 Å and Uiso(H) = 1.2 Ueq(C) for the carbon atom. The N-H and OH hydrogen atoms were located with a Fourier map with the distance for N-H bonds being 0.93 (3)Å to 0.98 (3)Å and O-H = 0.82Å to 0.85Å. The Uiso(H) was set to 1.2 Ueq(N) for nitrogen atoms and 1.5 Ueq(O) for oxygen atoms. The crystal data were deposited at the Cambridge Crystallographic Data Centre, and the office assigned it a CCDC number, which is 1,989,095. Crystal data can be received free of charge on application to CCDC 12 Union Road, Cambridge CB21 EZ, UK. (Fax: +44-1223-336-033; e-mail: data_request@ccdc.cam.ac.uk).

### 2.7. Computational Study

The DFT level of theory has been employed to carry out geometry optimization with the help of the Gaussian-16 [22] using the B3LYP functional [23]. For all other atoms, the computations were conducted with the 6-31G** basis set [24]. During the geometry optimization process, no symmetry constraints were used. The Hessian matrix was generated analytically for the optimized structures to validate the location of the right minima (no imaginary frequencies). Crystal Explorer [25] software was used to map the Hirshfeld surface using crystal structure data from CIF files.

## 3. Results and Discussion

### 3.1. Chemistry

The active methylene compound coumarin has a nucleophilic character and usually reacts with aldehydes and ketones to form dicoumarols. Biscoumarins, which are bridge-substituted dimers of 4-hydroxycoumarin, have also been discovered to have significant activity. The diverse applications of these derivatives have gained tremendous attention among synthetic chemists to develop an efficient synthetic route for these scaffolds. Among various biopolymers, chitosan stands as one of the most widely used supports for catalytic applications. Chitosan is synthesized by alkaline chitin deacetylation (Figure 1) [26]. The hydroxyl and amino group’s structure contributed to the chelating property and made it a useful chelating agent. Initially, the reaction of 4-hydroxycoumarin with 3-formylchromone was carried out using ethanol and piperidine as a solvent and catalyst, respectively, under reflux conditions. The reaction afforded the expected bis-adduct (**3**) in 4 h with a 68% yield [4,27]. However, when the reaction of 4-hydroxycoumarin with 3-formylchromone was carried out via chitosan catalyzed solvent-free green synthesis in the presence of piperidine in a molar ratio, it resulted a different product (**4**) from biscoumarin in 88% yield (Figure 2). The compound (**4**) is a proton transfer type complex and has cationic and anionic parts along with a water molecule, which produces a range of different hydrogen bonding interactions.

In order to optimize the reaction and to find the most appropriate condition for the specified reaction, different solvents were selected, such as PEG-600, ethanol, methanol, and water, and also tested in solvent-free conditions (Table 1). A comparative study shows that in ethanol, the reaction proceeded with a yield of 76% after 2.5 h. Similarly, methanol also took 3 h to complete the reaction with a 69% yield. The effect of water on the reaction gave comparable results with a 65% yield, like with ethanol and methanol, but took a long time (7 h) to complete. The result of PEG-400 is also unsatisfactory in terms of yield and time when compared to other solvents as well as solvent-free reactions. However, when the process was investigated in a solvent-free environment, the product **4** yield was improved significantly within shorter periods (Table 1), which highlights the importance of solvent-free reaction over the conventional solvent-assisted reaction. From this, we confirmed that the solvent-free condition was the best technique for novel product synthesis in terms of yield and time.

The increased surface area of the catalyst makes more active sites available for reactant interaction, resulting in a faster reaction rate. The presence of –OH and –NH groups of chitosan may provide more active basic sites for reaction and the formation of product in a better yield. The absence of solvents in solvent-free reactions may help to eliminate the dilution effect due to the solvent and the heat required as activation energy is directly made available to the reactant molecules. Besides, it prevents the chance of high collisions to form more than one product and wastes energy by heating the solvents. A plausible base-catalyzed mechanism for the formation of the product **4** has been given in Figure 3.

Satisfying the green chemistry criteria, the recyclability of a catalyst is an important requirement in catalysis. Thus, the reaction mixture was extracted with 10 mL of hot ethyl acetate after the reaction completion. The catalyst, which is insoluble in ethyl acetate left after extraction was washed again with ethyl acetate, dried in air, and used for the subsequent cycles. The results showed that there was no significant decrease in catalytic activity for up to four cycles (Table 2).

Spectral analysis established the incorporation of a piperidinium molecule in the initially formed dicoumarol unit. The structure of the new dicoumarol was established by the physical studies, IR and NMR. The carbonyl groups of coumarin and chromone units showed prominent absorption bands in infrared (IR) spectra at 1680 cm^−1^ and 1642 cm^−1^, respectively. The OH stretching frequency was discernible at 3412 cm^−1^. The ^1^H NMR spectra showed the piperidinium protons (1.946–3.676 ppm), besides the coumarin and chromone proton signals. Detailed signal patterns as observed are given in the experimental section. The ^13^C NMR spectrum also provided further confirmation of piperidinium carbons besides coumarin and chromone carbons. Further, the elemental analysis results were in concordance with the theoretical values.

### 3.2. Crystal Structure Description

In the crystal structure of title compound **4,** the central carbon atom is connected with three different ring systems, i.e., 4-hydroxy-2-oxo-2*H*-chromen (A), 4-oxo-4*H*-chromen (B) and 2-oxo-2*H*-chromen-4-olate (C) moieties, Figure 1. The dihedral angle between the planes of 4-hydroxy-2-oxo-2*H*-chromen (A) & 4-oxo-4*H*-chromen (B), 4-oxo-4*H*-chromen (B) & 2-oxo-2H-chromen-4-olate (C) and 4-hydroxy-2-oxo-2*H*-chromen (A) & 2-oxo-2*H*-chromen-4-olate (C) are 64.88(4)°, 84.51(4)° and 57.21(4)°, respectively. These are in comparison with the already reported crystal data for another molecule [26]. The piperidinium cation adopted the chair conformation with the root mean square deviation of 0.222(3) Å. The puckering parameters were observed from the PLATON and the values are provided here, *Q* = 0.5462, *θ* = 178.26° and *φ* = 6.66°. Table 3 shows the crystallographic parameters, whereas Appendix A include bond lengths and bond angles. The compound has a cation and anion along with a water molecule which produces a range of different hydrogen bonding interactions Table 4. The oxygen atom O7 acts as a donor via H1O to the O8 and generates interactions of intramolecular hydrogen bonding, which indicates a rise to eight-membered ring motif formation (O8/C4/C3/C1/C12/C13/O7/H1O) with the r.m.s. deviation of 0.310(3)Å where the most deviations were observed from C13 = 0.369(5)Å and C1 = 0.565(2)Å. A water molecule (H2O-O9-H3O) afforded four different hydrogen bonding interactions shown in Table 4, Appendix A, where O-H…O connects the chromone molecules, while N-H…O represents the interaction of the water molecule with the piperidine moiety. The piperidinium cation interacts via N-H and C-H bonds to the chromone and water molecules. Overall, these interactions generate three-dimensional interactions that stabilize the compound crystal structure.

### 3.3. Hirshfeld Surface Analyses

The Hirshfeld surface analysis is a helpful tool for understanding molecular crystal intermolecular interactions. Because of the size and form of the Hirshfeld surface, it is feasible to explore and observe both qualitative and quantitative intermolecular near interactions in molecular crystals. Crystal Explorer is used to build Hirshfeld surfaces and their associated fingerprint plots. Crystal Explorer 3.0 takes a crystallographic information file (CIF) as input. Furthermore, mapping normalized contact distance (dnorm) yields the identification of intermolecular interaction areas of significant value, which is expressed as:d_norm_ = (d_i_ − r_i_^vdw^)/r_i_^vdw^ + (d_e_ − r_e_^vdw^)/r_e_^vdw^
where r_i_^vdw^ and r_e_^vdw^ are the van der Waals radii of the atoms.

The interactions between the atoms within the crystal under research are once again described using a color gradient. This gradient changes from blue to crimson and then back to white. The bluish regions show that the distance between neighboring atoms exceeds the sum of their individual van der Waals radii when intermolecular interactions are taken into account. The white patches indicate locations where the distance between nearby atoms is close to the sum of the atoms’ van der Waals radius. The color red is used to show regions where the van fder Waals rays of surrounding atoms interpenetrate. The occurrence of non-covalent interactions between the atoms (or groups of atoms) located at the interface of the zones is represented in red, which exhibits a considerable approximation between these atoms, which is reasonable to suspect based on these facts. When the domains under consideration are illustrated in white, the situation becomes more delicate since the distance between adjoining atoms is at the limit of the sum of the van der Waals rays. Locations that are bluish indicate areas where neighboring atoms are too far apart to interact. Figure 2 shows the Hirshfeld surfaces of compound 4, which have been mapped over a dnorm range of −0.5 to 1.5, a shape index range of −1.0 to 1.0, and a curvedness range of −4.0 to 0.4. The parameter dnorm produced a surface with a red-white-blue color scheme, with deep red areas indicating shorter hydrogen bonding connections. The white patches represent contacts, such as H...H contacts surrounding the van der Waals separation, while the blue portions are free of such close contacts. Both red patches related to C-H... interactions and ‘bow-tie patterns’ indicating the presence of aromatic stacking (...) interactions can be seen on the Hirshfeld surface mapped with the shape index function. The electron density of surface curves around chemical interactions was illustrated by the curved surface.

Compound **4**’s fingerprint plots, shown in Figure 3, can be deconstructed to highlight specific atom-pair tight interactions. This itemization separates contributions from various interaction types that might otherwise overlap in the complete fingerprint. The proportions of H...H, H...O, C...H, C...C, and C...O interactions for each molecule are 34.9%, 30.7%, 25.6%, 5.2%, and 2.8% of the total Hirshfeld surface, respectively.

### 3.4. Density Functional Theory

The structural parameters and optimization of the molecule molecular structure in the ground state were computed using the Gaussian 09 package at the Becke3–Lee–Yang–Parr (B3LYP) level using the usual 6-31G** basis set. Figure 4 shows the optimized structure. The geometrical parameters, bond distances and angles are slightly higher than experimental results because theoretical calculations were performed assuming isolated molecules in the gaseous phase while geometry optimization was conducted utilizing initial coordination from solid crystals.

Figure 5A shows the FTIR signature of compound 4 measured in the 400–4000 cm^−1^ range compared to the computed IR spectrum. Because computed vibrational bands are subjected to systematic inaccuracies due to basis set truncation, poor treatment of electron correlation, and a harmonic approximation, the computed spectrum is regularly up-shifted by around 40–60 cm^−1^ compared to actual signals.

The -C=O symmetric stretching of the carbonyl group with -OH symmetric bend is ascribed to the most intense band in the region observed at 1680 cm^−1^ and estimated at 1780 cm^−1^. Ring deformation with C=C stretch and C-H bending located on the coumarin ring is detected at 1623 (DFT: 1644 cm^−1^) and 1593 cm^−1^ (DFT: 1613 cm^−1^). NH_2_ scissoring, which is anticipated by DFT at 1590 cm^−1^, has an extra low-intensity band in this region at 1577 cm^−1^. C-H (aliphatic and aromatic) vibrations were measured in the range 2800–3200 cm^−1^, and theoretically computed in the range 2890–3250 cm^−1^. At 3230 (DFT: 3296 cm^−1^) and 3287 cm^−1^ (DFT: 3315 cm^−1^), respectively, the N-H symmetrical and asymmetrical stretching vibrations were identified. The -OH stretching vibrations are attributed to a broad band in the range of 3350–3500 cm^−1^.

TDDFT computations were also used to characterize the electronic transitions for the experimentally discovered peaks. A complete set of TDDFT calculations were performed on the compound, at the B3LYP/ 6-311G* level of theory to characterize their singlet excited states and electronic properties. The UV-vis spectra obtained by TDDFT are in good agreement with the UV-vis spectra observed experimentally in Figure 5B. For compound 4, the absorbance band in the 250–350 nm region is characterized and dominated by π to π* electronic transition, and a major contribution is assigned to HOMO-1 → LUMO+1 excitation. The orbital compositions of computed singlet transitions are shown in Figure 6.

To further understand the nature of bonding and its reactivity, the molecule was investigated using the Kohn–Sham frontier molecular orbital (KS-FMO) method. The relative energies of HOMO energy levels are tremendously valuable, even though the exact values of these levels cannot be properly measured. Frontier orbital research, on the other hand, can provide a qualitative understanding of their bonding and reactivity. The HOMO-LUMO energy gap and a molecule’s chemical reactivity have been discovered to be linked. Chemical reactivity is linked to chemical hardness, which is defined as the resistance to perturbation in a molecule’s electrical charge distribution. When the KS-FMO is taken into consideration, chemical hardness is equal to the energy gap between HOMO and LUMO, which is approximated by (EHOMO−ELUMO)/2, where EHOMO and ELUMO are the energies of HOMO and LUMO, respectively. The larger the HOMO–LUMO energy gap, the harder it is for the electron cloud to deform, resulting in decreased reactivity. Similarly, because an electron cloud in such a setup is easily disturbed by an external field, the shorter the energy gap, the higher the reactivity. As a result, a wider gap indicates a lower level of response and vice versa. The energy difference between HOMO and LUMO has been calculated and found to be 3.90 eV (Figure 6b).

## 4. Conclusions

In summary, we have synthesized a novel piperidinium dicoumarol derivative by the reaction of 3-formylchromone, 4-hydroxycoumarin, and piperidine through an eco-friendly solvent-free greener approach using chitosan as a green catalyst. The present method was found to have excellent advantages in comparison to solvent-assisted conventional methods. The short time span, significant yields of products, and absence of toxic byproducts or wastes are highlighted advantages of the present green protocol. The structure of novel piperidinium dicoumarol has been established by FT-IR and NMR spectroscopy, which was further confirmed by a single X-ray diffraction study. The dicoumarol crystal structure was stabilized by hydrogen bond interactions, which was validated by Hirshfeld surface analysis. The structure of the dicoumarol derivative was further studied using density functional theory calculations with the B3LYP functional and 6-311++G(d,p) basis set. The simulated FT-IR and UV spectra were compared with the experimental ones and the discrepancies were discussed. This protocol can be efficiently applied to a wide variety of aldehydes as well as active methylene compounds in excellent yields. Moreover, solvent-free/solid-state synthesis makes this protocol more environmentally benign and a significant contribution toward sustainability. Therefore, it is expected that this experimental-computational work may provide new horizons for the generation of novel heterocycles having potential biological and industrial applications.

## Data Availability

Not applicable.

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
