# Peer review of "Chitosan Catalyzed Novel Piperidinium Dicoumarol: Green Synthesis, X-ray Diffraction, Hirshfeld Surface and DFT Studies"

_polymers, 2022, doi:10.3390/polym14091854_

Round 1

Reviewer 1 Report

The authors have described the synthesis of novel piperidinium dicoumarol using chitosan catalyzed solvent-free green conditions. The authors presented work as interesting and detailed experiments of green catalyst development, X-ray diffraction, Hirshfeld surface, and DFT studies. I would recommend considering this manuscript after clarifying the following critiques.

1.

It would be good if the authors investigated the synthesized compound's biological activity to improve the manuscript's strength.

2.

How could it help the temperature condition for the yield of products?

3.

Authors should also refer to the following recent article discussing the piperazine-based natural product hybrids. (Bioorganic Chemistry 82 (2019) 306–323; https://doi.org/10.1016/j.bioorg.2018.10.039).

4.

How could authors explain the stability of the piperidinium dicoumarol product?

5.

References have not been uniformly in the journal format. Make sure to correct it.

Author Response

uploaded

Reviewer 2 Report

Dear authors,
I read the article very carefully and after analyzing it I will recommend the publication with minor revisions.
Could the authors describe how they obtained the single crystal for X-ray diffraction?
I recommend that the authors explain the differences between compound 3 and 4.
In subchapters 2.3 and 2.4 the information is duplicated. Acknowledging the authors to check and leave in subchapter 2.4 only how they recovered the catalyst.
The article also requires a careful check of the English language used to write it.

Author Response

uploaded
